# Sensor-Based Human Activity Recognition Using Adaptive Class Hierarchy

**DOI:** 10.3390/s21227743

**Published:** 2021-11-21

**Authors:** Kazuma Kondo, Tatsuhito Hasegawa

**Affiliations:** Graduate School of Engineering, University of Fukui, Fukui 910-8507, Japan; t-hase@u-fukui.ac.jp

**Keywords:** human activity recognition, class hierarchy, deep learning

## Abstract

In sensor-based human activity recognition, many methods based on convolutional neural networks (CNNs) have been proposed. In the typical CNN-based activity recognition model, each class is treated independently of others. However, actual activity classes often have hierarchical relationships. It is important to consider an activity recognition model that uses the hierarchical relationship among classes to improve recognition performance. In image recognition, branch CNNs (B-CNNs) have been proposed for classification using class hierarchies. B-CNNs can easily perform classification using hand-crafted class hierarchies, but it is difficult to manually design an appropriate class hierarchy when the number of classes is large or there is little prior knowledge. Therefore, in our study, we propose a class hierarchy-adaptive B-CNN, which adds a method to the B-CNN for automatically constructing class hierarchies. Our method constructs the class hierarchy from training data automatically to effectively train the B-CNN without prior knowledge. We evaluated our method on several benchmark datasets for activity recognition. As a result, our method outperformed standard CNN models without considering the hierarchical relationship among classes. In addition, we confirmed that our method has performance comparable to a B-CNN model with a class hierarchy based on human prior knowledge.

## 1. Introduction

Human activity recognition is expected to be used in a wide range of fields [1]. Sensor-based human activity recognition is the task of automatically predicting a user’s activity and states using sensors. The prediction results can be used to support user actions or decision-making in organizations.

In recent years, deep learning (DL) has been used in various fields and many DL methods have been proposed for human activity recognition. Many activity recognition models with DL are based on convolutional neural networks (CNNs) [2]. DL is a powerful method for various fields and has been rapidly developed in computer vision and neural language processing especially.

The activity recognition models based on CNNs use activity labels encoded to one-hot vectors. Typical activity recognition models are trained ignoring the relationships among activities, because the one-hot encoding treats each class as independent of each other. However, there are hierarchical relationships among actual activities, which are based on similarity of sensor data [3]. For example, considering four classes of stationary, walking, going up the stairs and going down the stairs, the three classes other than stationary can be regarded as an abstract class, non-stationary. This indicates that there is a hierarchical structure among the activity classes. Hierarchical relationships among classes are known to affect classification patterns of standard CNNs [4]. Low similar classes, such as stationary and walking, can hardly be misclassified mutually. On the other hand, high similar classes, such as walking and going up the stairs, are frequently misclassified mutually. Therefore, the recognition model is expected to improve its performance using relationships among classes known in advance.

In some previous works on human activity recognition, hierarchical classification methods using class hierarchies have been proposed, as shown in Figure 1a. Hierarchical classification methods solve a decomposed classification problem based on a class hierarchy using multiple models. In this method, each model targets only the classification problem simplified by the class hierarchy, which reduces complexity and improves classification performance, compared to standard classification models. However, such hierarchical classification methods have the disadvantage of increasing the number of models when dealing with a large number of classification classes or classes with complex hierarchical relationships. Especially, in some hierarchical classification methods with DL [5,6], the increase in the number of models leads to a significant increase in computational costs.

On the other hand, in computer vision, Zhu et al. [7] proposed the branch CNN (B-CNN), which incorporates the hierarchical relationship among classes into the CNN model structure. As shown in Figure 1b, the B-CNN is designed to learn the hierarchical relationship between classes by mapping the CNN hierarchical structure to the class hierarchical structure. The B-CNN is a simple method for incorporating the class hierarchy into the CNN model structure and it is easy to use B-CNNs for activity recognition. Unlike hierarchical classification methods, B-CNNs can use the class hierarchy in a single model; however, the class hierarchy needs to be provided by humans as prior knowledge to construct branches.

Humans can construct a class hierarchy using a misclassification tendency of a typical classifier, a data similarity and a domain knowledge. It is easy to manually design a class hierarchy with a small number of target classes, but it is not easy to manually design a class hierarchy with a large number of target classes because of the complexity of the relationships among the classes. In this study, we propose a class hierarchy-adaptive B-CNN, which adaptively defines the class hierarchy used to train B-CNNs. Our method is shown in Figure 1c. The left side of Figure 1c shows a B-CNN and our method adds the method for automatically constructing a class hierarchy to a B-CNN. Our method automatically constructs a class hierarchy from training data and the B-CNN is trained using the constructed class hierarchy. In this study, we show that the B-CNN model is also effective in activity recognition and we address the problem of B-CNNs whereby the hierarchical structure of classes must be designed by humans.

The contributions of this study are as follows:We show the effectiveness of the B-CNN model in sensor-based activity recognition. In addition, by examining the effect of the number of subjects used for training on the recognition performance of the model, we find that the B-CNN model is particularly effective when the number of training data is small.By examining the relationship within the class hierarchy provided for the B-CNN and its recognition performance, we found that an inappropriate class hierarchy decreases the recognition performance of the model, indicating that a class hierarchy is an important factor that affects the performance of B-CNNs.The above verification also revealed that class hierarchies designed by humans are not always optimal.To construct class hierarchies that work effectively for B-CNNs, we propose a method for automatically constructing class hierarchies based on the distances among classes in the feature space and we demonstrate the method’s effectiveness.

## 2. Related Works

### 2.1. Human Activity Recognition

Human activity recognition is the task to recognize human activities and states using sensing devices, such as accelerometers and cameras. The solutions for human activity recognition are categorized into video-based methods [8,9] and sensor-based methods [1,2,8] based on the different sensing devices. In video-based activity recognition, RGB images and RGB-D images, which include the depth map, are used. The image data format can represent states of the whole body of a single person and multiple people as one data; thus, video-based human activity recognition is effective to recognize fine-grained activities (e.g., gestures) and activities of a crowd. However, it is difficult to constantly recognize the state of a specific user for video-based activity recognition using cameras. On the other hand, sensor-based human activity recognition methods use inertial sensors worn by users, such as accelerometers and gyroscopes on smartphones, or environmental sensors installed in a space. Especially, in activity recognition performed using mobile devices (e.g., smartphones and smartwatches), it is possible to recognize the activities of a specific user regardless of the surrounding environment, because the device always moves with the user. In this paper, we focus on sensor-based activity recognition using mobile devices such as smartphones.

In sensor-based activity recognition, many methods using DL have been proposed. Methods without DL perform activity classification using features manually designed by humans. On the other hand, methods using DL perform feature extraction from observed sensor data and activity classification simultaneously. The methods based on DL are powerful and generally achieve higher performance than methods without DL.

In activity recognition models using DL, many CNN-based methods have been proposed for activity recognition [10,11,12,13,14,15,16,17]. Most of them use a simple model structure consisting of several convolutional layers and pooling layers, which are connected hierarchically [10,11,12]. Other methods that combine CNNs and recurrent neural networks (RNNs) have also been proposed [13,14,15]. Ordóñez et al. [13] and Xu et al. [14] proposed methods that extract spatial features from sensor data using CNNs and then classify the activity by RNNs from the extracted features. On the other hand, the method proposed by Xia et al. [15] encodes the time dependency of waveform data using RNNs, then extracts the spatial features by CNNs and classifies the activity by fully connected layers. Recently, Gao et al. [16] and Ma et al. [17] proposed an activity recognition method using the attention mechanism, which has attracted considerable attention in the fields of natural language processing and computer vision.

### 2.2. Usage of Class Hierarchy in Human Activity Recognition

In activity recognition, many hierarchical classification methods using a class hierarchy have been proposed [5,6,18,19,20,21]. The method proposed by Khan et al. classifies activities to three abstract classes—stationary, non-stationary and transition—and then classifies target classes included in each abstract class. Their methods achieved much better performance than models without hierarchical classification. Fazli et al. [5] and Cho et al. [6] proposed hierarchical classification methods using DL models such as MLP and CNN. Their method achieved higher performance than standard DL models. However, in terms of the use of class hierarchies, their methods only replace the models used in the previous hierarchical classification methods with DL models.

Van Kasteren et al. [22] proposed an activity recognition method with the hierarchical hidden Markov model (HHMM), which uses relationships among classes. Their method assumes that an activity consists of some actions (e.g., raising an arm and putting a foot forward) and the actions are latent variables, which are not annotated. The actions are detected automatically, such as a clustering, in the process of optimizing the model. Their method outperformed a hidden Markov model and a hidden semi-Markov model without the actions.

### 2.3. Usage of Class Hierarchy in Computer Vision

In the field of image recognition, Zhu et al. [7] proposed the B-CNN, which reflects a class hierarchy in the structure of the CNN. B-CNNs classify coarse classes based on a class hierarchy in a branching path in the middle of the model. This enables the model to learn the hierarchical relationships among classes, thereby improving the recognition performance of the model. Similar to B-CNNs, various methods have been proposed to reflect the hierarchical structure of classes in CNN models [23,24,25,26]. On the other hand, Deng et al. [27] proposed the hierarchy and exclusion (HEX) graph that represents the hierarchical relationships among classes and a classification method using the HEX graph. Unlike B-CNNs, their method encodes the hierarchical relationship between classes using the HEX graph and trains the recognition model that is based on conditional random fields. Koo et al. [28] proposed the method for capturing the hierarchical relationship among classes using RNNs. Their method extracts hierarchical features of images using CNNs and predicts the class hierarchy using RNNs.

The B-CNN is a very simple method to learn hierarchical relationships among classes in CNNs. In the Zhu et al.’s work, a B-CNN was verified using only one type of class hierarchy that they designed. However, the effect of structures of the class hierarhcy on B-CNN performance was not discussed in their work.

### 2.4. Automatically Constructing Class Hierarchy

Most of the methods mentioned above assume that the class hierarchy is manually designed based on the human’s prior knowledge. On the other hand, there have been approaches to automatically construct class hierarchies from data. Methods for constructing class hierarchies are mainly divided into top-down methods [29,30,31,32,33] and bottom-up methods [33,34,35,36,37]. In the top-down method, a virtual abstract class is treated as a root node of a tree structure and a class hierarchy is constructed by recursively partitioning this root node until all classes become leaf nodes. In the top-down method, structures of class hierarchies are less constrained because the nodes can be divided in arbitrary numbers. However, the number of node divisions is a hyperparameter and must be set by humans. On the other hand, in the bottom-up method, each class is merged recursively with certain criteria until all classes are merged into one abstract class. Concretely, the bottom-up methods using hierarchical clustering [34,36], confusion matrices [33] and graphs [37] have been proposed. Hierarchical clustering creates dendrograms by recursively merging two classes (or clusters) based on the distances between features. The created dendrogram can be regarded as a kind of class hierarchy, but its structure is constrained by the binary tree. By extracting flat clusters from the dendrogram using a threshold, the constraint of the structure can be relaxed to some extent. However, the threshold needs to be given by humans. Our method determines the threshold for the dendrogram automatically. Hence, our method can construct class hierarchies without human’s prior knowledge at all.

The work by Jin et al. [37] is particularly relevant to our study. Their method detects confusion communities from a graph called confusion graph using the Louvain method [38], which is one of the community detection algorithms. The confusion graph is created from the softmax outputs of CNNs. Their method can construct class hierarchies because the Louvain method can output a process of detecting communities.

### 2.5. Hierarchical Multi-Label Classification

Hierarchical multi-label classification (HMC) is another task that deals with hierarchical relationships among classes. HMC is a task aimed at inferring hierarchical relationships among classes [39,40] and is known to be more complex and difficult than typical classification tasks because HMC involves inference of multiple classes as well as hierarchical relationships among classes. Although HMC is similar in concept to the automatic construction of class hierarchies, they have different objectives. In automatic construction methods for class hierarchies, the quality of the hierarchical structure is not the main target of evaluation, since the hierarchical structure is composed on the assumption that the class hierarchy will be used for another task. On the other hand, in HMC, the quality of the estimated hierarchical structure is the main evaluation target of the method since the objective is to estimate the hierarchical structure of classes.

This research study aims to improve the performance of a specific activity classification problem using a hierarchical structure of classes; therefore, it differs from HMC.

## 3. Class Hierarchy-Adaptive B-CNN Model

In this section, we describe B-CNNs and our automatic class hierarchy construction method, which are the main components of our method.

Figure 1c shows our proposed method. Our method classifies activities by B-CNNs using a class hierarchy automatically constructed from training data. Our method consists of two steps, as described in Algorithm 1. In our method, feature vectors are firstly computed from a pre-trained standard CNN model. Next, the centroids of each class in the feature space are calculated from the feature vectors and the class hierarchy is constructed by merging similar classes through hierarchical clustering. Then, the B-CNN is trained using the constructed class hierarchy. Our method uses two different models, $M_{std}$ and $M_{branch}$. $M_{std}$ is a standard CNN model used for class hierarchy construction, and $M_{branch}$ is a B-CNN model. $M_{branch}$ is trained from the initial state without using the parameters of the trained $M_{std}$.
**Algorithm 1** Class hierarchy-adaptive B-CNN.**Input:** Train Dataset for B-CNN $D_{train}={\left\{\left(x_{i},y_{i}\right)\right\}}_{i=0}^{N-1}$; Split rate $r_{pre}$; Dimension for PCA *d*; The number of corase levels in class hierarchy *L*;
**Output:** Trained B-CNN Model $M_{branch}$
1:// (1) Construct Class Hierarchy (see Algorithm 2)2:$P\leftarrow \mathrm{ConstructClassHierarchy}(D_{train},r_{pre},d,L)$3:$P^{\left(0\right)},P^{\left(1\right)},\cdots ,P^{\left(L-1\right)}\leftarrow P$4: 5:// (2) Train B-CNN Model6:Initialize B-CNN $M_{branch}$7:let be $p_{i}^{\left(k\right)}\in P^{\left(k\right)}\left(i=0,\cdots ,N-1,k=0,1,\cdots ,L-1\right)$.8:$D_{train}^{branch}\leftarrow {\left\{\left(x_{i},p_{i}^{\left(0\right)},p_{i}^{\left(1\right)},\cdots ,p_{i}^{\left(L-1\right)},y_{i}\right)\right\}}_{i=0}^{N-1}$9:Train $M_{branch}$ using $D_{train}^{branch}$10:**return** $M_{branch}$


Although we propose a class hierarchy-adaptive B-CNN for sensor-based activity recognition, our method can be applied to video-based activity recognition based on CNNs, such as the models designed by Ji et al. [41] and Zhou et al. [42]. Our method can be applied to a variety of other problem settings. To apply the method, the following three conditions are required:The task can be formulated as a classification problem and the estimation target can be grouped into abstract concepts.The classification deep learning model is composed of stacked convolution layers, such as VGG and ResNet.The entire model can be pre-trained in an end-to-end manner such as training with softmax cross-entropy loss to design a class hierarchy by our method.

In this section, Section 3.1 describes the B-CNN proposed by Zhu et al. [7] and Section 3.2 shows our automatic class hierarchy construction method.

### 3.1. Branch Convolutional Neural Network (B-CNN)

The structure of the model is shown in Figure 1b. The B-CNN branches a model into multiple paths based on a class hierarchy and classifies them in ascending order from the highest level in the class hierarchy. Similar to traditional CNN models, the B-CNN classifies using class scores calculated using the softmax function and each level of classification is performed independently.

The B-CNN is trained using the stochastic gradient descent method. The loss function is a weighted sum of the softmax cross-entropy loss of each level and the loss function $L_{i}$ for the *i*-th sample is defined in Equation (Equation 1):(1)$$L_{i}=-\sum_{k=1}^{K}\sum_{c=1}^{C_{k}}w_{k}t_{i,c}^{k}\log\frac{e^{y_{i,c}^{k}}}{\sum_{j=1}^{C_{k}}e^{y_{i,j}^{k}}}$$
where *K* denotes the number of levels in the class hierarchy; $w_{k}$ represents the weight for the loss of the *k*-th level; $C_{k}$ represents the number of classes in the *k*-th level; $y_{*,c}^{k}$ denotes the classification score of class *c* in the *k*-th level; $t^{k}$ represents the ground truth in the *k*-th level and is defined in Equation (Equation 2).
(2)$$t_{*,c}^{k}=\left\{\begin{array}{cc} 1 & \left(c\;\mathrm{is}\;\mathrm{ground}\;\mathrm{truth}\right)\\ 0 & \left(\mathrm{otherwise}\right) \end{array}\right.$$

In the B-CNN model, a convolutional block consisting of several convolutional layers and a pooling layer is used as the unit of the branching position. Since a typical CNN model has a structure in which multiple convolutional blocks are connected hierarchically, various patterns of branching are possible. In this study, the patterns of the branching positions are tuned by treating them as a hyperparameter.

### 3.2. Automatic Construction of Class Hierarchies

The procedure for creating the class hierarchy is shown in Algorithm 2. Figure 2 shows the method for automatically constructing a class hierarchy.
**Algorithm 2** Constructing a class hierarchy.**Input:** Training Dataset for B-CNN $D_{train}$; Split rate $r_{pre}$; Dimension for PCA *d*; Number of target classes *C*; The number of corase levels in class hierarchy *L*;
**Output:** Hierarchical multi-labels based on constructed class hierarchy, *P*
1:// (1) Pre-training phase2:Initialize Standard CNN $M_{std}$3:$D_{pre},D_{adp}\leftarrow \mathrm{SplitDataset}\left(D_{train},r_{pre}\right)$4:Train $M_{std}$ using $D_{pre}$5:$F\leftarrow \mathrm{ExtractFeature}(M_{std},D_{adp})$6:F←DecompositionWithPCA(F,d)7: 8:// (2) Calculate centroids for each target class9:Let $F^{center}$ be an array of size *C*.10:**for** c=0,⋯,C−1**do**11:    $N_{c}\leftarrow \mathrm{The}\;\mathrm{number}\;\mathrm{of}\;\mathrm{samples}\;\mathrm{in}\;\mathrm{class}\;c$12:    $F_{c}\leftarrow \left\{f_{c}\in F;f_{c}\mathrm{belonging}\;\mathrm{to}\;\mathrm{class}\;c\right\}$13:    $F^{center}\left[c\right]\leftarrow \frac{1}{N_{c}}\sum_{f_{c}\in F_{c}}f_{c}$14:**end for**15: 16:// (3) Construct class hierarchy17:$H\leftarrow \mathrm{CreateDendrogram}(F^{center})$18:let d0,d1,⋯,dk be the distances between the clusters in the order of integration in the dendrogram *H*.19:*k* denotes the number of clusters in dendrogram *H*.20:$s_{i}:=d_{i+1}-d_{i}\;\left(i=0,1,\ldots ,L-1\right)$21:// $m_{i}$ denotes a mapping from target label to corse label.22:// $P^{\left(i\right)}$ denotes abstract class labels of target classes.23:let *Y* be target class labels included in $D_{train}$.24:**for each** i=0,1,⋯,L−1**do**25:    $j\leftarrow \underset{i}{\mathrm{ArgMax}}\left(S\right)$26:    $t_{i}\leftarrow d_{j}+\epsilon \;\left(\epsilon >0\right)$27:    $m_{i}\leftarrow \mathrm{HierarchicalClustering}\left(H,t_{i}\right)$28:    $P^{\left(i\right)}\leftarrow \mathrm{Relabe}\left(Y,m_{i}\right)$29:    Pop $s_{j}$ from *S*30:**end for**31:$P\leftarrow {\left\{P^{\left(i\right)}\right\}}_{i=0}^{L-1}$32:**return** *P*


In the creation of the class hierarchy, first, the standard CNN model $M_{std}$, which has no branch structure, is trained with $D_{pre}$, while $D_{adp}$ is converted into a feature vector *F* by the $M_{std}$. Then, *F* is reduced to *d* dimensions using the principal component analysis (PCA). In this study, the number of dimensions was set to d=64. $D_{pre}$ and $D_{adp}$ are created by dividing the $D_{train}$ used for training the B-CNN model $M_{branch}$. The $r_{pre}$ is the ratio of $D_{pre}$ to $D_{train}$ and, in this study, $r_{pre}$ was set to $r_{pre}=0.5$. Next, the class centroids $F^{center}$ are calculated from the feature vectors *F*. $F^{center}$ is treated as a representative vectors of each class. Then, a dendrogram *H* is created by performing a hierarchical cluster analysis on $F^{center}$. There are various types of hierarchical cluster analyses. In this study, we used the Ward method [43].

To create a class hierarchy, a threshold is determined based on the distance between the merged clusters. In the hierarchical cluster analysis, two data (or clusters) are integrated recursively according to certain criteria and, finally, all data are integrated into one cluster. The dendrogram shows the order of the clusters to be integrated and the distance between the clusters in merging them. In Figure 2, class 4 and class 5 are first merged as one cluster; then, class 3 and cluster {4, 5} are merged. In our method, the distances between the merged clusters are defined as $d_{0},d_{1},d_{2},\cdots ,d_{k}$ in the order of merging and $s_{i}=d_{i}-d_{i-1}\;\left(i=1,2,\cdots ,k\right)$ is calculated. Then, L elements are selected from $S={\left\{s_{i}\right\}}_{i=0}^{k-1}$ and, for each selected element $d_{j}$, $d_{j}+\epsilon$ is calculated and set as the threshold. Finally, coarse labels are determined by clustering using these thresholds and a class hierarchy is constructed from the coarse labels and target labels.

## 4. Experimental Settings

In our experiment, we evaluated the estimation accuracy using three different benchmark datasets. In this section, we describe the details of the experimental setups.

### 4.1. Dataset

In the experiments, three datasets were used: HASC [44], WISDM [45] and UniMib SHAR [46].

HASC [44] is a benchmark dataset for which basic human activities were recorded by wearable devices such as smartphones. HASC contains sensor data for six types of activities: stay, walk, jog, skip, stair up (stup) and stair down (stdown). Sensors used in HASC include accelerometers, gyroscopes and magnetic sensors; however, in this study, we used only a 3-axis accelerometer. In the experiment, BasicActivity data with a sampling frequency of 100 Hz were used and the position and type of the device were not restricted. We trimmed 5 s before and after each measurement file. The input data for the model were created using the sliding window method. The window size and stride width were set to 256 samples. We used the data of 176 subjects, which account for a sufficient amount of data among all subjects recorded in HASC.

WISDM [45] is a benchmark dataset containing data relative to human daily activities measured using smartphones, as well as HASC. In WISDM, 3-axis accelerometer data are recorded, relative to six types of activities: standing (stand), sitting (sit), walking (walk), jogging (jog), ascending stairs (stup) and descending stairs (stdown). The WISDM holds records of activities similar to HASC, but the dataset contains fewer subjects and a smaller amount of data than HASC. In our experiments, the accelerometer data were divided into several segments based on subjects and activities and trimmed 3 s before and after each segment. The input data for the model were created using the sliding window method; the window size and stride width were set to 256 samples.

UniMib SHAR [46] is a dataset that contains measurements of human daily activities and fall scenes using smartphone accelerometers. The measured daily activities and fall scenes are standing up from sitting (standFS), standing up from lying (standFL), walking (walk), running (jog), jumping (jump), going upstairs (stup), going downstairs (stdown), lying down from standing (layFS), sitting (sit), generic falling forward (fallF), generic falling backward (fallB), falling rightward (fallR), falling leftward (fallL), hitting an obstacle in the fall (hitO), falling with protection strategies (fallPS), falling backward–sitting-chair (fallBSC) and syncope (syncope). UniMib holds records of 3-axis accelerometer data and is provided in a frame of 151 samples. The experiment was conducted using the divided data.

### 4.2. Model Structure

In our method, two types of models are used, the standard CNN model (we call this model the std model) and the B-CNN model. In this study, the VGG model [47] is used as the base model for both models. The VGG model has been confirmed to be effective in sensor-based activity recognition by Hasegawa et al. [48] and we judged that the simple model structure of VGG is appropriate as the base model for the B-CNN. Figure 3 shows the structure of the B-CNN model used in the experiments. Conv Block consists of several convolution layers and a max pooling layer, and Classifiers consist of a global average pooling layer and one fully connected layer. In the VGG model proposed by Simonyan et al. [47], the feature map obtained from the last convolutional layer is resized to 7×7 using average pooling and pooled features are input to a classifier consisting of several fully connected layers and a dropout layer. In this study, we aim to verify the effect of the B-CNN on the feature space and the resulting change in recognition performance, so the size of the classifier was kept small to reduce the effect of the classifier on the model performance. In addition, the original VGG model uses two-dimensional convolutional layers to handle images as input, but, in this study, all convolutional layers were changed to one-dimensional (1D) convolutional layers to handle 1D sensor data as input. The branching position of the B-CNN was tuned for each dataset as a hyperparameter. In Figure 3, the Conv Blocks are 1, 2, 3, 4 and 5 from the left, while the branching positions are 1 and 3 for HASC, 1 and 2 for WISDM and 3 and 4 for UniMib.

### 4.3. Training Model

All models were trained using Adam and the initial learning rate was set to 0.001. The number of epochs was set to 300, but 200 epochs were set for the training of the std model $M_{std}$, which was used for automatic construction of class hierarchy. When training the model, three types of data augmentation methods were used together: RICAP [49], sensor data axis swapping and sensor data amplitude inversion. Although RICAP is a method proposed in the field of image recognition, it can be applied to 1D data such as sensor data and its effectiveness has been confirmed by Hasegawa [50], along with the rotation of sensor data. RICAP has a hyperparameter β, which is related to the cut-out position of the data. In this study, we set β=0.5. The inversion of the sensor data amplitude can expand the distribution of input data with respect to the terminal position and orientation as well as the axis swapping and is considered to be effective in datasets such as HASC, where the terminal possession position and orientation are not fixed.

In addition, in the training of the B-CNN, it is necessary to set weights $w_{k}$ for each loss on the hierarchy. In this study, we set the weights of each hierarchy equally, with the constraint that the sum of the weights of each hierarchy was one. We focused on the effects of differences in class hierarchies and set the weights in this manner to minimize the effects of the weights $w_{k}$.

### 4.4. Evaluating Model

The hold-out method was used to evaluate the model and the dataset was divided by subjects. The breakdown of the number of subjects included in Train (training set), Validation (validation set) and Test (test set) is shown in Table 1.

The HASC dataset includes large-scale sensor data from a large number of subjects—more than 100. Therefore, we set the number of subjects in the training, validation and test sets to 10, 50 and 50, respectively, to ensure a sufficient number of data and variety in the validation and test sets for the HASC’s validation. In contrast, in WISDM and UniMib SHAR, the number of subjects was set the same, so that all the data would be used in each trial, because they have a small number of subjects compared to HASC. This is because the same partitioning method as that used for HASC may not provide enough data for the validation and test sets.

In the experiment, a set of 20 trials was conducted, consisting of division of the dataset, training of the model and evaluation of the model; the average of the results of all trials was used for evaluation. The indices used for evaluation were accuracy and average F-score.

## 5. Experimental Results

Table 2 shows the accuracy and average F-score of our method and the baseline method. In the table, stdvgg16 represents the standard CNN model with the VGG structure and branchvgg16 represents the CNN model with the B-CNN branch added to the VGG structure. In this section, we first present the validation results on the effectiveness of the B-CNN model and then discuss the results. Afterward, we discuss the effects of different class hierarchies on the recognition performance of the B-CNN model and the search costs of class hierarchies. Then, based on the above two points, we evaluate the effectiveness of our method and analyze the class hierarchies created using our method.

### 5.1. Discussion on the Effectiveness of B-CNNs

#### 5.1.1. Effectiveness of B-CNNs

Table 2 shows the experimental results of the std model and the B-CNN model on each dataset. For B-CNN, we compared three methods for creating the class hierarchy: the hand-crafted method, Jin et al.’s [37] method and our method. In the hand-crafted method, a B-CNN is trained using class hierarchies that are manually designed by humans. The hand-crafted class hierarchy is described in detail in Section 5.3.

Table 2 shows that the B-CNN model with the manually designed class hierarchy outperformed the std CNN model in terms of F-score on all datasets. In UniMib, the B-CNN model is inferior to the std model in terms of accuracy, but the difference is as small as 0.002. Therefore, the B-CNN model is not only effective in image recognition but also sensor-based activity recognition. Furthermore, Figure 4 shows the change in accuracy when the number of subjects in $D_{train}$ was increased by 10, from 10 to 50, in the HASC dataset. According to the results of the B-CNN with the manually designed class hierarchy, the B-CNN model outperformed the std model in terms of accuracy, even when the number of subjects was increased. However, as the number of subjects used for training increased, the difference in accuracy between the B-CNN and std models decreased. Thus, the B-CNN model is a particularly effective method when the number of training data is small.

#### 5.1.2. A Study on the Effect of Backbone Architecture on the Recognition Performance of B-CNNs

B-CNNs can use any CNN architecture as the backbone. In the work by Zhu et al. [7], a B-CNN with a VGG-like architecture as the backbone was used. However, their work did not discuss the effects of backbone CNNs on the recognition performance of B-CNNs. Therefore, in our work, we also examined the effects of the architecture and size of the backbone of B-CNNs on recognition performance.

In the validation, we compared B-CNNs with three different backbone architectures: VGG [47], ResNet [51] and LSTM-CNN [15]. In the VGG architecture, we also compared four different models with different depths and different widths (the number of filters). The architecture of the LSTM-CNN was composed of simple CNNs following the two LSTMs, while the VGG16 was used as the simple CNN. The class hierarchy used in the B-CNN was designed manually. The same class hierarchy was used for all models. The branching position of the B-CNN was tuned for each architecture, which were VGG, ResNet and LSTM-CNN. Each model was validated using the HASC dataset using the same method as described in Section 4.4.

The validation results are shown in Table 3. In the table, VGG16-S represents the VGG16 model whose convolution filters were all halved and VGG16-W represents the VGG16 model whose convolution filters were all doubled. As the results show, in the VGG architecture, the accuracies of all models with a branching structure were higher than those of the models without a branching structure. The VGG16-S with a branching structure achieved the highest recognition performance, but the gain of recognition performance due to branching structure was the largest for the plain VGG16. The VGG19, which was deeper than the VGG16, and the VGG16-W, which was wider than the VGG16, had smaller improvements in recognition performance than the plain VGG16 due to the branching structure.

The recognition performance of ResNet and VGG models without branching structure was the same. However, in the ResNet architecture, unlike VGG, there was no performance improvement due to the branching structure. The most significant difference between ResNet and VGG was the presence of a skip connection. Therefore, it is considered that B-CNNs are less effective in architectures with the skip connection.

The recognition performance of the LSTM-CNN model without the branching structure was higher than the VGG and ResNet models without the branching structure. The recognition performance of the LSTM-CNN model was improved by using the branching structure. In addition, the recognition performance of LSTM-CNNs with branching structure and VGG16 with branching structure were equivalent. This result implies that the B-CNN has a particularly large effect on the CNN part of the LSTM-CNN and a small effect on the LSTM part.

#### 5.1.3. A Study on the Effect of Different Class Hierarchies on the Recognition Performance of B-CNNs

We examine the relationship within the class hierarchy provided for B-CNNs and the recognition performance of the model. All possible class hierarchy creation patterns were attempted using the HASC dataset. Since the number of class hierarchies was extremely large (3230 patterns), we train and evaluated the model once for each class hierarchy and treated the evaluation result as the score of the class hierarchy.

Figure 5 shows a histogram of the accuracy of all trials. The blue vertical bar represents the frequency of each class and the red line represents the cumulative relative frequency. This result shows that the accuracies of all trials were distributed around 0.808 and the trials with an accuracy greater than or equal to 0.808 accounted for 50% of the total trials. Since the accuracy of the std model was 0.805, it is likely to achieve higher accuracy than the std model, even if the class hierarchy is created randomly. The minimum and maximum accuracies of the B-CNN model were 0.771 and 0.838, respectively. The recognition performance of B-CNNs varies significantly depending on the design of the class hierarchy.

Table 4 shows the class hierarchies with the maximum and minimum accuracy in all trials, respectively. In the class hierarchy with the highest accuracy, stay and skip were integrated in Level 2, whereas stay, skip, walk and skip were integrated in Level 1. Figure 6 shows the percentage of integration of two different classes in Level 2 of the top 1% accuracy class hierarchy. According to this result, in the Level 2 hierarchy of the top 1%, the three classes walk, stup and stdown had the largest percentage of integration with each other. Furthermore, the percentage of integration between skip and stay, which was seen in the class hierarchy with the highest accuracy, was large. In Level 1, the percentage of two classes merged increased, compared to Level 2, but the trend was generally the same as in Level 2. When humans manually design class hierarchies, they consider stay and skip to be the classes with the least similarity to each other and assign them to different clusters, as well as assigning stay and a group of walk and stup to different clusters as stationary and non-stationary. This shows that the class hierarchies designed by humans are not necessarily the best class hierarchies for B-CNNs.

On the other hand, in the class hierarchy with the lowest accuracy, stay and jog, walk and stdown, and skip and stup were integrated in Level 2, while stay, jog, walk and stdown were integrated as one cluster in Level 1. In Figure 7, the percentage of integration of two different classes in Level 2 of the lower 1% accuracy class hierarchy is shown. According to the results, the percentage of integration of stay and jog, stay and stup, skip and stup, and skip and stdown was large in the Level 2 hierarchy of the bottom 1% and, especially, stay and stup is a pattern that also appears in the lowest class hierarchy in Table 4. This trend is similar to the Level 1 hierarchy. Since walk, stup and stdown are classes that are frequently misclassified with respect to each other, the integration of stay and stup in the lowest class hierarchy may have negatively affected the classification of walk, stup and stdown in the target classes, degrading the classification accuracy. In contrast, in the pattern where stay and skip are integrated, which is often seen in the top 1% accuracy class hierarchy, stay and skip are completely different activity classes. Therefore, it is considered that, even if a B-CNN model is optimized with stay and skip as the same class in the branched classifier, sufficient feature representation for activity classification can be obtained by optimizing the target classes.

#### 5.1.4. Search Costs of Class Hierarchy

In Section 5.1.1, we verify that B-CNN improves activity recognition performance. However, in Section 5.1.3, it is shown that inappropriate class hierarchy degrades the recognition performance of the B-CNN model. Therefore, the design of the class hierarchy is an important factor of B-CNNs. One method for designing the class hierarchy is to manually design a class hierarchy based on humans’ prior knowledge. In this method, it is difficult to manually design the class hierarchy when the number of original classes is large. Here, Figure 8 shows the number of patterns created for a class hierarchy of height 3, as shown in Figure 2. According to the graph, the number of patterns of the class hierarchy increases exponentially as the number of the original classes increases. For example, when the original number of classes is 4, the number of patterns for creating a class hierarchy is 18. But, when the number of classes is 9, the number of patterns of the class hierarchy is as large as 7,226,538 and all searches are unrealistic.

Furthermore, the class hierarchy needs to be created considering a relationship among classes. For example, in the case of UniMib, the 17 activity classes can be semantically divided into two classes, daily activities and fall scenes. However, the activities of “lying down from a standing (layFS)” included in the daily activities and “falling backward (fallB)” included in the fall scenes may be similar in terms of body movements, regardless of whether they are falls or not. Thus, when manually designing a class hierarchy, it is necessary to consider the similarity of each activity in the analysis, which makes the manual design of a class hierarchy an extremely difficult task when there are many classes.

From the above discussion, a method for automatically designing a class hierarchy from data is useful, especially when the number of classes is large.

In this study, we use an automatic design of class hierarchies for training B-CNNs, but it can be also useful in interpreting the target task itself through the automatically designed class hierarchy.

### 5.2. Discussion of the Proposed Method for Automatic Construction of Class Hierarchies

In this subsection, based on the above discussion, we evaluate the effectiveness of our method. Table 2 shows that our method outperforms the std model in both accuracy and F-score for all datasets. In addition, our method outperforms the B-CNN model with class hierarchies constructed by Jin et al.’s method [37]. Comparing our method with the B-CNN model using manually designed class hierarchies, our method achieved a classification accuracy comparable to the model using manually designed class hierarchies for both accuracy and F-score metrics on HASC and WISDM. In UniMib, our method had the highest accuracy, but its F-score was 0.006 lower than that of the B-CNN model with a manually designed class hierarchy. Furthermore, Figure 4 shows that our method outperformed the std model in terms of accuracy, even when the number of subjects used for training was increased; however, the difference in accuracy with the std model became smaller as the number of subjects increased. Our method had slightly lower accuracy than the B-CNN using the manually designed class hierarchy, regardless of the number of subjects used for training, but the difference became smaller as the number of subjects increased. Therefore, the effect of different class hierarchies on the B-CNN model was more pronounced when the number of training data was small.

These results show that class hierarchies manually designed by humans based on prior knowledge work well for B-CNNs. However, the performance of our method is comparable to that of the B-CNN model using manually designed class hierarchies. Therefore, our method is particularly effective when it is difficult to design class hierarchies that work well for B-CNNs with little prior knowledge.

### 5.3. Discussion on Class Hierarchy Designed Using the Proposed Method

We show the class hierarchies designed using our method in detail and evaluate them qualitatively. In Table 5 and Table 6, class hierarchies manually designed for each dataset and class hierarchies automatically designed using our method for a particular trial are shown. Table 6 shows the class hierarchy for the UniMib dataset, but, because the UniMib dataset has a large number of classes, the table is wrapped in the middle. In the table, Level 1 and Level 2 correspond to the coarse classes in Figure 1c and Level 3 corresponds to the target classes.

According to the results of HASC, only walk and stup were integrated in Level 2, which is a reasonable result in terms of activity similarity. However, in Level 1, stay, walk, stup and stdown were integrated as one class. Even in other trials, there were many cases in which stay, walk, stup and stdown were integrated into one class. However, considering the similarity in the activities, the stay and walk groups (walk, stup and stdown) should be separated. Therefore, based on the discussion in Section 5.1.3, it can be considered that this is the reason for the difference in recognition performance between our method and the B-CNN model with the manually designed class hierarchy.

In addition, it is thought that one of the reasons why the stay and walk groups (walk, stup and stdown) were merged into one class is the loss function used to train the std model. In our method, the distribution of each class in the feature space was designed by training the std model and the class hierarchy was created using the distribution of each class in the feature space. The designed feature space depends on the loss function that is optimized in training the std model and it is thought that the data of the walk group were distributed closer to the data of the stay than the data of the skip on the feature space designed by optimizing softmax cross-entropy loss used in this study. Therefore, it may be possible to create a more appropriate class hierarchy by examining the loss function used for training the std model and the training method for the std model.

In WISDM, the result for walk differs significantly from that of the manually designed class hierarchy. In the manually designed class hierarchy, walk was integrated into the same cluster as stup and stdown at Level 2 and was integrated with jog, stup and stdown at Level 1 in terms of the magnitude of the motion; however, in the class hierarchy created by our method, walk was not integrated with any class at Level 2 and was integrated with jog at Level 1. Both jog and walk move on a flat surface and their movements are similar, even though the magnitudes of the movements are different. It is considered that our method integrated walk and jog into one cluster and separated them from stup and stdown because of this similarity.

In UniMib, standFS, standFL, layFS and sit are stationary activities such as standing up and sitting down, whereas fallF, fallPS, fallR, fallL, fallB, fallBSC, hitO and syncope are grouped into one activity, classed as fall motion. It is difficult to distinguish among walk, jog, stup, jump and stdown, but it is considered that they are differentiated by the magnitude of body movement. stup and stdown are similar activities, but stdown is different from stup in that it moves toward gravity. Therefore, in stdown, the body moves faster than in stup when going up and down a staircase and the acceleration observed by the sensor becomes larger. This is a common term with jump, which has a large acceleration when jumping up, and it is thought that our method separated walk, jog and stup from jump and stdown by the magnitude of acceleration.

From the above discussion, it can be seen that, although the class hierarchies created using our method were different from that designed by a human, many of the class hierarchies were designed by capturing the similarity of activity from a perspective different from that of humans. Table 2 also shows that the B-CNN model trained with class hierarchies created using our method achieved better recognition performance than the std model. However, based on the discussion in Section 5.1.3, there may be more appropriate class hierarchies that can improve the recognition performance of B-CNN models and there is still room for improvement in our method.

## 6. Conclusions

In this study, we proposed a class hierarchy-adaptive B-CNN model for human activity recognition. Our method automatically creates a class hierarchy from the training data and trains the B-CNN using the created class hierarchy. Thus, our method performs classification considering the hierarchical relationships among classes without prior knowledge. The experimental results show that the B-CNN model is also effective for sensor-based activity recognition. In addition, we found that the B-CNN model is particularly effective when the amount of training data is small. Next, we evaluated our method and confirmed that our method achieves better classification accuracy than the standard CNN model and achieves a recognition performance comparable to the B-CNN model with a manually designed class hierarchy. Since the search costs of class hierarchies required for B-CNN training increase exponentially with the number of classes, our method is particularly effective in situations where there is little prior knowledge and there is difficulty in creating an appropriate class hierarchy. However, the class hierarchies created using our method depend heavily on the feature space designed by training the standard model and, if the hierarchical relationship among classes is not reflected in this feature space, it is difficult to design appropriate class hierarchies using our method. In addition, our results suggest that there is a class hierarchy that improves activity recognition accuracy more than the class hierarchy automatically created using our method. Therefore, the design of the features used to create the class hierarchies and the methods for creating more appropriate class hierarchies from the features are future tasks. 

## Figures and Tables

**Figure 1 sensors-21-07743-f001:**
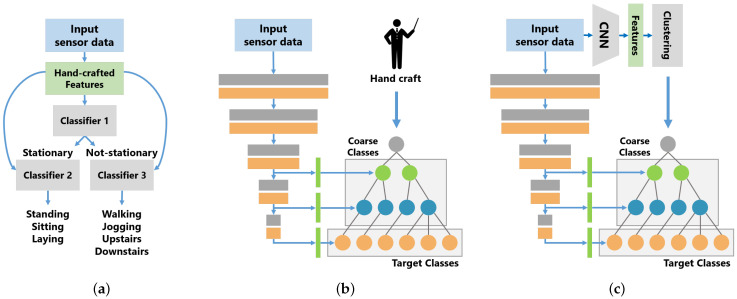
Model overview of the related methods (**a**,**b**) and of our method (**c**). (**a**) Hierarchical Classification. (**b**) B-CNN [7]. (**c**) Class Hierarchy Adaptive B-CNN.

**Figure 2 sensors-21-07743-f002:**
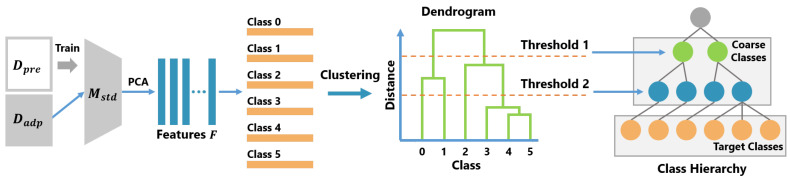
Overview of the class hierarchy construction method.

**Figure 3 sensors-21-07743-f003:**
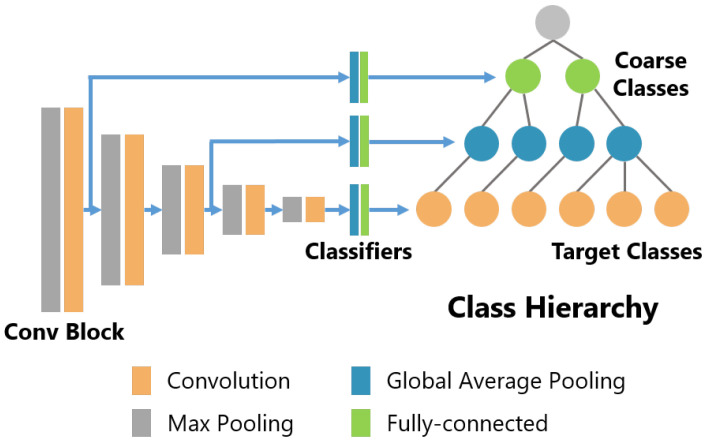
Model structure of the B-CNN with VGG16 as the base model.

**Figure 4 sensors-21-07743-f004:**
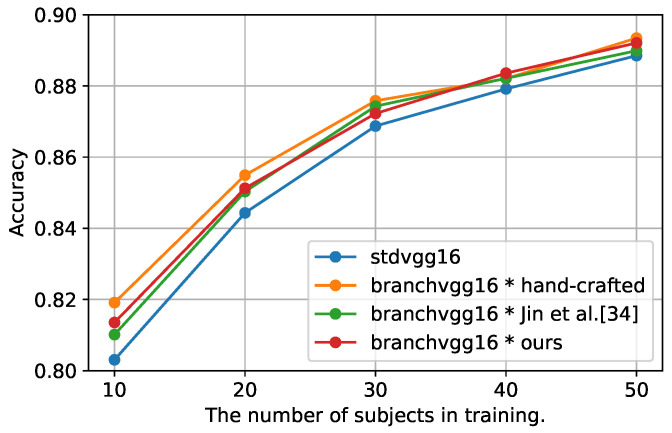
Effect of varying the number of subjects used for training on accuracy.

**Figure 5 sensors-21-07743-f005:**
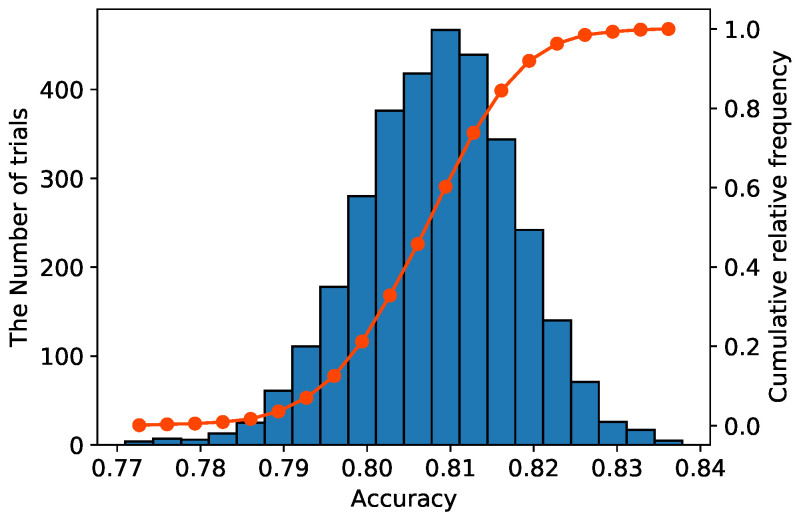
Distribution of accuracy when attempting all possible patterns of class hierarchy in the HASC dataset.

**Figure 6 sensors-21-07743-f006:**
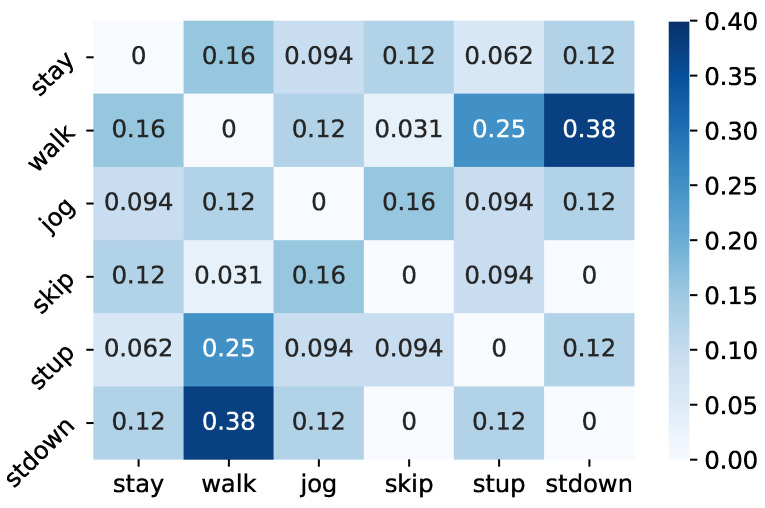
Percentage of two classes merged at Level 2 of the top 1% class hierarchy.

**Figure 7 sensors-21-07743-f007:**
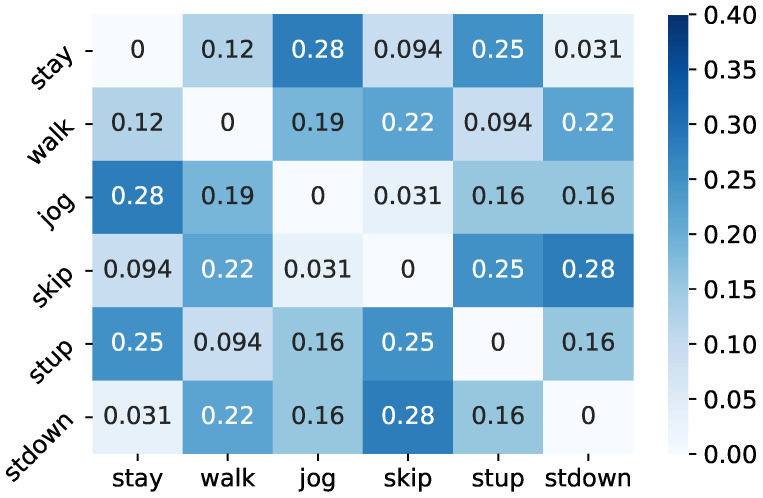
Percentage of two classes merged in Level 2 of the lower 1% class hierarchy.

**Figure 8 sensors-21-07743-f008:**
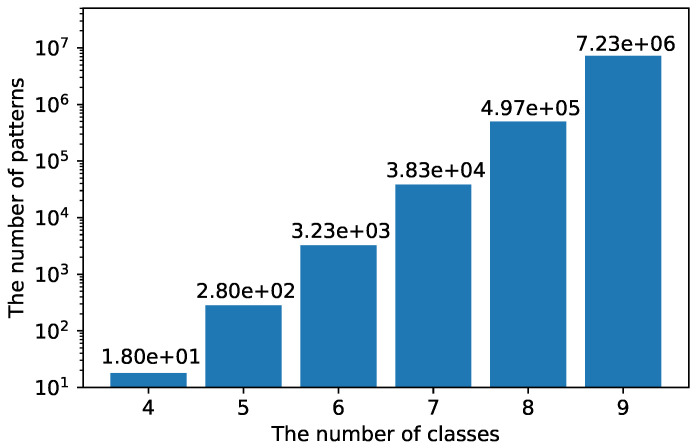
The number of combinations of class hierarchies of height 3 per original number of classes.

**Table 1 sensors-21-07743-t001:** Details of the number of subjects in each dataset in the hold-out method.

Dataset	Train	Validation	Test
HASC	10	50	50
WISDM	12	12	12
UniMib SHAR	10	10	10

**Table 2 sensors-21-07743-t002:** Comparison of estimation accuracy for three types of activity recognition datasets.

Model	Class Hierarchy	Branch	HASC		WISDM		UniMib	
			Accuracy	F-Score	Accuracy	F-Score	Accuracy	F-Score
stdvgg16	-	-	0.803	0.806	0.866	0.799	0.725	0.607
branchvgg16	hand-crafted	✓	**0.819**	**0.823**	**0.887**	**0.830**	0.723	**0.620**
branchvgg16	Jin et al. [37]	✓	0.810	0.814	0.870	0.801	0.719	0.607
branchvgg16	ours	✓	0.814	0.817	0.881	0.827	**0.728**	0.614

**Table 3 sensors-21-07743-t003:** Comparison of estimation accuracy for B-CNNs of different backbone architecture.

Backbone	w/o Branch		w/ Branch	
Architecture	Accuracy	F-Score	Accuracy	F-Score
VGG11	0.802	0.806	**0.816**	**0.820**
VGG13	0.808	0.812	**0.821**	**0.824**
VGG16	0.803	0.806	**0.819**	**0.823**
VGG16-S	0.811	0.815	**0.821**	**0.825**
VGG16-W	0.802	0.805	**0.811**	**0.814**
VGG19	0.801	0.805	**0.808**	**0.811**
ResNet18	**0.809**	0.809	0.807	**0.811**
ResNet50	0.797	0.799	**0.798**	**0.801**
LSTM-CNN	0.815	0.818	**0.820**	**0.824**

**Table 4 sensors-21-07743-t004:** Class hierarchy with maximum and minimum accuracy in all trials.

Level	HASC Best Hierarchy					
1	stay	skip	walk	stup	jog	stdown
2	stay	skip	walk	stup	jog	stdown
3	stay	skip	walk	stup	jog	stdown
	**HASC Worst Hierarchy**					
1	stay	jog	walk	stdown	skip	stup
2	stay	jog	walk	stdown	skip	stup
3	stay	jog	walk	stdown	skip	stup

**Table 5 sensors-21-07743-t005:** Manually designed class hierarchies for HASC and WISDM and class hierarchies designed by the proposed method.

(a) HASC												
**Level**	**Hand-Crafted**						**Proposed Method**					
1	stay	walk	stup	stdown	jog	skip	stay	walk	stup	stdown	jog	skip
2	stay	walk	stup	stdown	jog	skip	stay	walk	stup	stdown	jog	skip
3	stay	walk	stup	stdown	jog	skip	stay	walk	stup	stdown	jog	skip
(**b**) **WISDM**												
**Level**	**Hand-Crafted**						**Proposed Method**					
1	jog	walk	stup	stdown	sit	stand	jog	walk	stup	stdown	sit	stand
2	jog	walk	stup	stdown	sit	stand	jog	walk	stup	stdown	sit	stand
3	jog	walk	stup	stdown	sit	stand	jog	walk	stup	stdown	sit	stand

**Table 6 sensors-21-07743-t006:** Manually designed class hierarchy for UniMib SHAR and class hierarchy designed by the proposed method.

Level	Hand-Crafted Method									
1	jog	walk	stup	stdown	jump	standFS	standFL			
2	jog	walk	stup	stdown	jump	standFS	standFL			
3	jog	walk	stup	stdown	jump	standFS	standFL			
1	layFS	sit	fallF	fallPS	fallR	fallL	fallB	fallBSC	hitO	syncope
2	layFS	sit	fallF	fallPS	fallR	fallL	fallB	fallBSC	hitO	syncope
3	layFS	sit	fallF	fallPS	fallR	fallL	fallB	fallBSC	hitO	syncope
**Level**	**Proposed Method**									
1	jog	walk	stup	stdown	jump	standFS	standFL	layFS	sit	
2	jog	walk	stup	stdown	jump	standFS	standFL	layFS	sit	
3	jog	walk	stup	stdown	jump	standFS	standFL	layFS	sit	
1	fallF	fallPS	fallR	fallL	fallB	fallBSC	hitO	syncope		
2	fallF	fallPS	fallR	fallL	fallB	fallBSC	hitO	syncope		
3	fallF	fallPS	fallR	fallL	fallB	fallBSC	hitO	syncope		

## Data Availability

The data used to support the findings of this study are available from the corresponding author upon request.

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
