# Peer review of "Sensor-Based Human Activity Recognition Using Adaptive Class Hierarchy"

_sensors, 2021, doi:10.3390/s21227743_

Round 1
Reviewer 1 Report
This paper investigates adaptive class hierarchy in training of CNN model for sensor-based activity recognition. The authors proposed a method to automatically create a class hierarchy from the training data and train the B-CNN using the created class hierarchy. Their method performs classification considering the hierarchical relationship between classes without prior knowledge. The experimental results on three datasets indicated the effectiveness of the B-CNN model. Overall, the paper is clearly motivated and is technically sound. Minor concerns are:
1. Can the the proposed solution be adapted to video-based action recognition? If yes, what are the differences compared to those solutions for video-based action recognition?
2. Can the proposed model provide an interpretable prediction? please clarify. (see for example: "Visually interpretable representation learning for depression recognition from facial images," IEEE TAC, 2020.)
3. While the paper focuses on the sensor-based action recognition, it is necessary to cite and discuss some recent works on video-based solutions with class hierarchy.
Reviewer 2 Report
The comments are as follows,
- In 4.4 Evaluating Model, the authors use three datasets. Why the partition of HASC is different from WISDM and UniMib SHAR?
- 2 shows the basic model structure you use. But why the authors use 5 max pooling layers and convolution layers with 3 classifier? Is it the best setting for all datasets? The authors may add more experiments to show how the parameters of model affect the results, for example change the number of convolution layers.
- In related work, there many deep learning methods to solve this problem. The authors might add more baselines to compare with your model.
Round 2
Reviewer 1 Report
I have no further comments.